# Unlocking insights: Mahout's perceptions and practices in managing Elephant Endotheliotropic Herpesvirus (EEHV) infection among captive Asian elephants in Surin province, Thailand

Narueporn Kittisirikul[1], Nuttapon Bangkaew[2], Waraphon Phimpraphai[3], Supaphen Sripiboon[4]*

1 Graduate School, Faculty of Veterinary Medicine, Kasetsart University, Bangkok, Thailand, 2 Elephant Kingdom Project, Zoological Park Organization of Thailand, Surin, Thailand, 3 Department of Veterinary Public Health, Faculty of Veterinary Medicine, Kasetsart University, Bangkok, Thailand, 4 Department of Large Animals and Wildlife Clinical Science, Faculty of Veterinary Medicine, Kasetsart University, Bangkok, Thailand

* ssripiboon@gmail.com

**Data Availability Statement:** All relevant data are within the paper and its Supporting Information files.

## Abstract

Surin, situated in the northeastern region of Thailand, has earned the reputation of being an "*elephant village*" due to its high captive elephant population and unique tradition of elephant rearing. However, the continuous occurrence of elephant endotheliotropic herpesvirus (EEHV) infection poses a significant threat to elephants, particularly the young ones. This study investigated various aspects of EEHV-related elephant care among ninety-two mahouts at the Surin Elephant Kingdom Project. This study used semi-structured interviews and observations to assess the mahouts' knowledge, attitude, and practice (KAP scores) toward EEHV transmission, prevention, and management. The result revealed knowledge and practice scores below expectations, indicating an insufficient understanding the nature of disease and preventive measures. However, the mahouts exhibited awareness of the severity of the disease and factors contributing to transmission risk. Regarding the relationship among KAP scores, a positive correlation was observed at a low level (p < 0.05) between the knowledge and practice scores. Interestingly, approximately 55% of the survey respondents were confident that their elephants would not receive EEHV, leading to inadequate prevention measures. From the result, it is crucial to provide comprehensive knowledge about the nature of the disease and preventive measures to all mahouts. This education should emphasize the importance of early monitoring signs, appropriate weaning age, and preventing viral transmission practices. The KAP survey offers valuable insights that can identify areas requiring improvement and guide the development of effective and targeted disease prevention programs within the specific population. Therefore, it is recommended that the KAP survey should be employed in other parts of the country where the elephant management system differs.

**Funding:** This research benefited from the support of two funding sources. Narueporn Kittisirikul received the Kasetsart Veterinary Development Funds grant (number 64_03), which was employed for both data collection and publication purposes. Furthermore, she obtained partial funding from the Faculty of Veterinary Medicine, Kasetsart University (grant number 2563-1/02), which assisted in covering master's degree tuition fees and providing salary support.

**Competing interests:** The authors have declared that no competing interests exist.

## Introduction

In Thailand, an estimated 3000 Asian elephants (*Elephas maximus*) are kept in captivity for various purposes, including zoo education, tourist activities, rescue centers, and traditional blessing ceremonies [1–4]. These captive elephants are distributed throughout the country, with the province of Surin in northeastern Thailand having the largest population among all [5]. The Suay or Kuy people are local to Surin and have historical expertise in capturing and training wild elephants, resulting in the development of a unique culture of captive elephant handling [6]. Although capturing wild elephants is currently prohibited, this elephant-handling skill has been passed down through generations and continues to be practiced in the present day [7]. Traditional elephant-raising practices in Surin differ from those of other tourist elephant camps. Captive elephants in Surin are privately owned by residents and are being raised and kept close to the household, with family members regarding them as part of their own family [8]. For this instance, elephants are generally held in groups of families or females.

The most renowned location for captive elephants in Surin is the Ban Ta Klang village, also known as the "*elephant village.*" The Elephant Kingdom Project (EKP) was initiated in Ban Ta Klang village in 2009 to rescue elephants that previously worked in tourist businesses or street begging and bring them back to the town [9]. The EKP also aims to generate employment opportunities through ecotourism while preserving the traditional relationship between the local community and their elephants. Elephant owners within the EKP still retain ownership of their elephants, but they are required to raise the elephants within the designated area and refrain from taking them to work at tourist camps and/or street begging. Additionally, owners have the freedom to organize activities for their elephant within the EKP areas and allow them to attend traditional ceremony in Surin province, which lead to the interaction among between elephants. Due to the close contact between elephant-elephant and elephant-human in this area, the potential for disease transmission through direct and indirect contact among elephants and humans could result in a widespread epidemic if appropriate disease surveillance and preventive measures are not implemented.

In the field of zoo and wild animal medicine, the threat of elephant endotheliotropic herpesvirus was recently prioritized. EEHV is a fatal hemorrhagic disease in elephant calves, which also has been documented in Thailand, including Surin [10, 11]. Addition, a total of 11 confirmed EEHV cases was reported in Surin with 66.7% fatality rate from 2006–2019 [11, 12]. This disease is responsible for high case fatality rate of 69% and an even more elevated fatality rate of 80–85% once clinical symptoms are presented [11–13]. Early clinical indications of EEHV hemorrhagic disease are non-specific, including lethargy, anorexia, changes in sleeping patterns, fever, and gastrointestinal symptoms such as colic and diarrhea [11]. EEHV-specific signs generally emerge rapidly in the advanced stages of the disease, including facial edema, tongue cyanosis, and bloody diarrhea [11, 12, 14]. Elephants calves between the ages of 1 and 8 years are considered at higher risk for EEHV hemorrhagic disease [15, 16]. This fatal disease poses a significant threat to the entire captive elephant population in Thailand, with 40 reported elephant deaths attributed to this disease between the years 2006–2019 [11]. Treatment with antiviral drugs is not always conclusive [17], with its efficacy being most significant when administered in the early stages of infection [12]. Therefore, due to the efficiency of early antiviral drug medicine coming from early detection, it is recommended that weekly real-time PCR monitoring should be conducted in susceptible groups of elephants [12]. However, limitations such as funding, the wide distribution of each elephant, logistics, and laboratory capacity may impede the feasibility of routine laboratory monitoring in most areas of Thailand. As an alternative, raising awareness, improving knowledge, and understanding of the disease among elephant caretakers could be a practical and effective means of monitoring for signs of

EEHV and implementing preventive measures, thereby reducing the morbidity and mortality associated with EEHV infection in the population.

A knowledge, attitude, and practice (KAP) survey are a research tool designed to assess the key components of a specific population concerning a particular topic of interest. KAP surveys are commonly used in public health to provide valuable information for resource allocation and implementation of public health programs [18]. One of the goals of KAP research is to identify gaps in knowledge, negative attitudes, or unhealthy practices to design and implement effective interventions. This field of study can also assess the extent of disease awareness, as demonstrated by Lungten et al. (2021), who reported an improvement in rabies prevention and safety knowledge levels following an awareness education program [19]. Recently, KAP surveys have been used to assess the understanding of the influencing factors on COVID-19 prevention and control. The results showed deviations in residents' understanding and knowledge, leading to adopting relevant preventive measures [20]. Moreover, the results from KAP surveys can be combined with inferential statistics to identify the significant factors affecting knowledge, attitudes, and practices, as demonstrated in a recent study of the KAP survey of COVID-19 among undergraduate students in China [21].

This current study employed a KAP survey to identify the specific needs and challenges for developing effective interventions for EEHV infection in Surin. This survey was assessed during the COVID-19 pandemic when the movement of elephants was higher due to the closing of elephant tourism activities. The primary focus of this study was on the mahouts, who were responsible for managing and caring for elephants. Previous research has highlighted the mahouts' significant role as a valuable source of knowledge about captive Asian elephant [22]. Their practices showed the effect to elephant behavior [23] and health [24]. Therefore, mahouts are considered a critical stakeholder in elephant management and prevention. The insights gained from this study can help develop targeted educational programs and communication strategies to improve EEHV prevention and control.

## Materials and methods

### Study area

In December 2021, cross-sectional KAP surveys was performed at the EKP in Thea Tum district, Surin province (latitude, longitude: 15.254010, 103.490556). The EKP held 200 registered elephants. However, several elephants were found to inhabit the adjacent areas, named Ban Tatip, Ban Taklang, Ban Kapho, and Ban Sala.

### Sample size

The knowledge, attitude, and practice information regarding EEHV were collected from mahouts (elephant caretakers) using two main methods: interviews from the semi-structured questionnaire and field observation. The sample size was calculated using the Taro Yamane method [25]; initially, purposive sampling under the inclusion criteria that all mahouts with elephants aged less than ten years old were first selected (n = 41), then simple random sampling was used for sampling the rest of the calculated sample size (n = 51).

### Data collection methods

Participants were interviewed following semi-structured questionnaires by a trained interviewer to avoid bias and increase accuracy. The questionnaire was divided into five sections with a total of 21 questions. *Section 1* contained five questions regarding demographic information (gender, age, number of elephants under care, years of experience with elephants, and

type of management). *Section 2* had three questions about elephant transportation, cleaning, and quarantine protocol. *Section 3* included four short answer questions regarding basic knowledge of EEHV infection, including the disease's common name, clinical signs, mode of transmission, and prevention. In addition, *Section 4* contained eight closed-ended questions related to the participant's attitudes toward the risk of EEHV transmission, including their awareness of EEHV's severity, treatment, and agreement of EEHV transmission from various risk factors. In section 4, participants were asked to rate their level of agreement on a Likert scale (agree, unsure, disagree, and not answered). Lastly, *Section 5* was about the practices according to the EEHV's prevention and management, which contained a single open-ended question: how do mahouts prevent their elephants from becoming infected with EEHV?. In addition, the field observation was applied to assess husbandry, waste management, and general care provided to elephants, in comparison to insights gathered through interviews.

The KAP scores were measured from three sections (*sections 3 to 5*). Section 3 evaluated the participants' knowledge of EEHV infection and was subdivided into three topics. Each topic encompassed an understanding of clinical signs, routes of transmission, and disease prevention, with a maximum of 10 points for each subject. For the knowledge of clinical signs, participants were able to earn up to five points for recognizing early symptoms (such as depression, anorexia, decreased appetite, reduced activity, and fever), by receiving one point for each sign for a maximum of five points. An additional five points were receiving from identifying any of specific EEHV-related symptoms (such as organ swelling (eyelid, neck, tongue, body, face), tongue cyanosis, bloody diarrhea, and petechial hemorrhage on the tongue). In the assessment of knowledge on transmission routes, six points were allocated for elephant-to-elephant direct contact, recognized as the primary route of transmission. An extra four points were assigned for any other form of transmission, which constituted as indirect transmission. Within the knowledge on EEHV prevention, participants were able to score up to 10 points, with seven points designated for specific practices aimed at preventing EEHV infection and controlling its spread, while additional three points were allocated for demonstrating good practices related to maintain overall health condition and general disease prevention.

Attitude scores were gauged through a Likert scale featuring three levels of agreement: agree (2 points), unsure (1 point), and disagree/not answer (0 points). Individuals who posed awareness of disease severity, understanding in disease transmission, and maintain a positive attitude toward EEHV prevention would receive higher points. In the context of EEHV-related practices, four points were initially allocated for general practices promoting good health and related to preventive medicine in general. An additional point, with a maximum of six points, was granted for specific practices aimed at EEHV prevention. These practices encompassed closely monitoring in behavior changes, isolation of EEHV infected case, refraining from contact, or sharing items with non-quarantine elephants.

Study site survey and the general data collection for the development of questionnaires took place from October 4, 2021 to November 4, 2021. On-site interviews were conducted from December 27 to December 31, 2021. Interviews were conducted in Thai language, and questions were adjusted to be easily understood, following a structured question framework. Data collection through observation methods persisted until the end of January 2022. Participants were apprised of the consent procedures by a facilitator and duly signed the consent form before the commencement of the interviews. In instances involving individuals under the age of 18, the consent form encompassed the signatures of both the parent or guardian and the participant. This study was approved by the Kasetsart University Research Ethic Committee (Approval no. KUREC-SS64/231).

## Statistical analysis

The results were analyzed and transformed into descriptive statistics, reported as numbers, percentages, and data visualizations. The inferential statistics was conducted using the R program (version 4.1.0) [26]. The statistical analysis included correlation testing to examine the relationships between knowledge, attitude, and practice scores, and the mahouts' demographics (such as age and experience in elephant handling). Additionally, the Wilcoxon test was used to compare KAP scores between mahouts who care for elephant calves and those who care for adult elephants. While the Mann-Whitney U test was used to compare KAP scores between the experience in EEHV infected elephant groups and the non-experience groups.

# Results

## General characteristics of respondents

From the total of 200 mahouts in EKP, this study included 92 participants (mahouts) from 81 households. The majority of the participants were male (85%, 78/92). On average, the participants were 41.6 years old, ranging from 14 to 70 years. Participants had an average of 25.6 years of experience handling elephants, ranging from 2 months to 63 years.

In this study area, the common practice was to assign one specific mahout to each elephant. In addition, each elephant typically had a primary mahout who majority interact with the elephant, other family member could assist in some practices, but would not count as mahout. On average, each family cared for 2–3 elephants, but the number of elephants per family ranged from one to eight. As observed in the field, most elephants were raised alongside their mother and siblings. The elephants were usually tethered close to each other, with their trunks touching (Fig 1). About 90% of the participants raised their elephants near areas where people lived using either a galvanized roof stall (Fig 2A) or under the shade of a tree (Fig 2B). In comparison, 10% tied the elephants with a long chain in a field-shaded area. Additional details about the participants are presented in Table 1.

## Mahout's knowledge toward EEHV

The interview was begun by directly asking "Do you know herpesvirus in elephants?". It was noted that participants recognized herpesvirus in elephant (EEHV) by various names. In addition, more than half of the respondents (56.52%, 52/92) were familiar with the general term "Herpesvirus", while one-third referred to it as "Cor-teep" in Thai language, which referring to neck stiffness and difficulty swallowing, which was the general term for diphtheritic disease. Additionally, 10.86% (n = 10/92) had never heard of or received information about EEHV before. Information about EEHV in the area was obtained from veterinarians, observing cases, community members, and promotional posters, respectively.

The mean knowledge score (MKS) among mahouts was 5.71 ± 3.96 (Table 2), ranging from 0 to 17. Less than half of the respondents (48.91%, 45/92) scored below the MKS. Overall, the participants demonstrated limited knowledge, with all aspects of knowledge scoring less than 50% of the total score, particularly in disease prevention.

In this study, data collection revealed 24 symptoms related to EEHV answered by the participants, which could be categorized into early non-specific symptoms, severe EEHV-specific symptoms, and mistaken symptoms. The most frequently mentioned symptom was lethargy (34.78%, 32/92), followed by a swollen face (27.17%, 25/92) and edema in various body organs (19.57%, 16/92), respectively. There were 11 misconceptions regarding clinical signs of EEHV, such as signs related to the upper gastrointestinal tract and swallowing problems. Furthermore, 14.13% (13/92) of the participants could not identify any EEHV clinical signs.

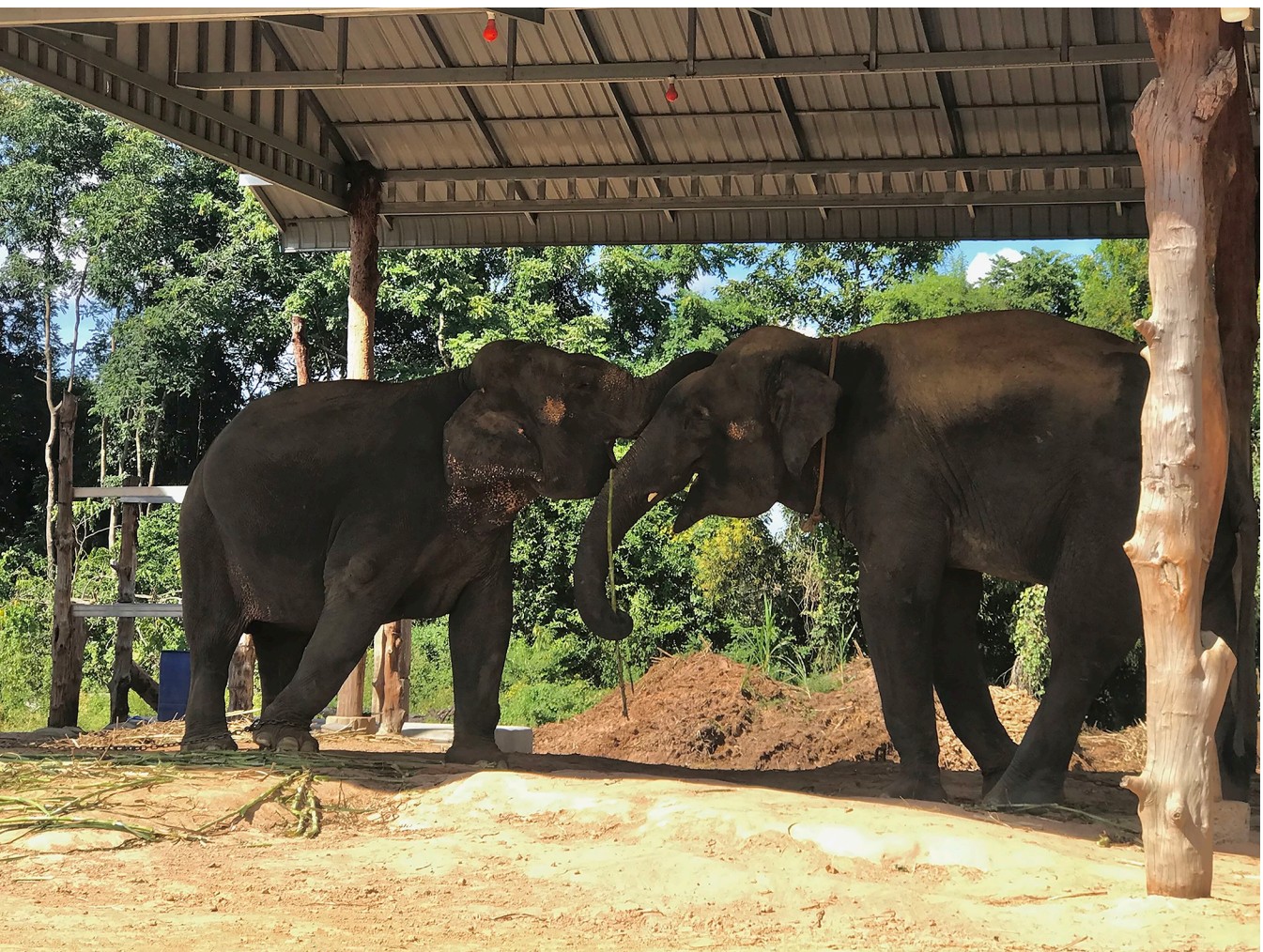

**Fig 1. Elephant were tethered close to one another at a trunk-to-trunk distance.**

For the knowledge regarding to the transmission route, only 39.13% (36/92) of the participants knew that EEHV could be transmitted through elephant-to-elephant direct contact, which was identified as the main transmission route. The second most mentioned transmission route was contacting the saliva from EEHV-infected cases, following the feeding with EEHV-infected elephants. However, 45.65% (42/92) of participants did not know any possible transmission routes.

Regarding to EEHV prevention knowledge, approximately half of the participants (48.91%, 45/92) knew that separating sick elephants from healthy ones could prevent disease transmission, and 33.70% (31/92) recognized that close monitoring for early clinical signs could improve treatment effectiveness. Giving daily vitamin C as the immune stimulator (7.61%; 7/92) and limited number of people in contact with sick elephants (3.26%; 3/91) were also stated for disease prevention. However, more than half of the participants (55.43%, 51/92) lacked knowledge of preventing or controlling EEHV infection.

## Mahout's attitude and perception toward EEHV infection

The mean attitude score (MAS) among mahouts in this area was 9.67 ± 0.47 (Table 3), with scores ranging from 0 to 14. Approximately two-thirds of the respondents (64.13%, 59/92)

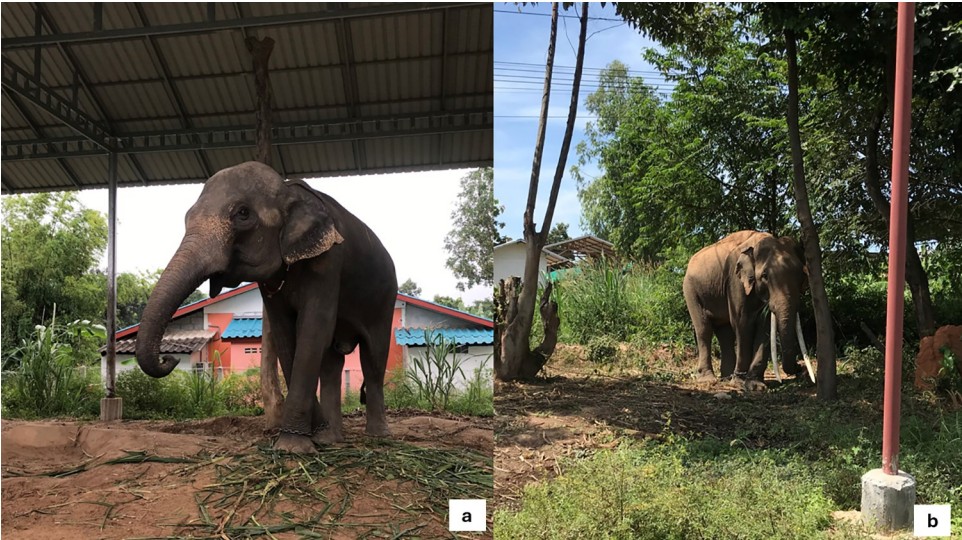

**Fig 2.** Two types of elephants resting area in this study sites, galvanized roof stall (a) and under the tree shad (b).

agreed on the severity of the disease. Respondents recognized risk factors associated with EEHV infection, including the susceptibility of hosts based on age, the fatality rate of infected cases, and the importance of vitamin C supplementation. However, there was disagreement regarding the risk of EEHV infection associated with newly introduced elephants and early weaning practices.

**Table 1. Demographics and general characteristics of respondents.**

| Variables | Response | Frequency | Percentage | Range |
|---|---|---|---|---|
| **Gender** | Male | 78 | 84.78 | - |
| | Female | 14 | 15.22 | |
| **Age (years)** | ≤ 15 | 3 | 3% | 14–70 |
| | 16–30 | 12 | 13% | |
| | 31–45 | 44 | 48% | |
| | 46–60 | 25 | 27% | |
| | > 60 | 8 | 9% | |
| **Number elephant under care** | 1 | 25 | 27% | 1–8 |
| | 2–3 | 39 | 42% | |
| | 4–5 | 18 | 20% | |
| | > 5 | 10 | 11% | |
| **Experiences with elephants (years)** | ≤ 5 | 8 | 9% | 2 months to 63 years |
| | 6–10 | 6 | 7% | |
| | 11–20 | 26 | 28% | |
| | 21–30 | 21 | 23% | |
| | 31–40 | 21 | 23% | |
| | 41–50 | 6 | 7% | |
| | 51–60 | 3 | 3% | |
| | > 60 | 1 | 1% | |
| **Management system** | Mahout's home space rearing area | 83 | 90.21 | - |
| | Forest area | 9 | 9.78 | |

**Table 2. Knowledge score regarding to EEHV infection.**

| Knowledge score | Total score | Mean ± SD | Range | number of participant's score ≥ mean (%) |
|---|---|---|---|---|
| Route of transmission | 10 | 2.38 ± 2.48 | 0–8 | 44 (47.83%) |
| Clinical sign | 10 | 2.02 ± 1.37 | 0–6 | 33 (35.87%) |
| Disease prevention | 10 | 1.31 ± 1.42 | 0–6 | 30 (32.61%) |
| **Total** | **30** | **5.71 ± 3.96** | **0–17** | **45 (48.91%)** |

Approximately 35% (32/92) of respondents suggested that elephants should be weaned at the age of two to three years old. Despite the concerns about early weaning which could interfering protective immune development, 55.81% (24/43) of the participants still weaned their calves for training before two years old. During this survey, participants expressed confidence in their traditional methods of raising elephants, such as grouping elephants, providing exercise, and allowing them to freely browse for food in the forested area with a long chain. Respondents believed these practices effectively reduced stress levels and promoted the overall health of the elephants. As a result, 55.43% (51/92) of the respondents expressed confidence that their elephants were unlikely to be infected with EEHV.

## Mahout's practice toward EEHV prevention and control

The participants' mean practice score (MPS) was 2.30 ± 1.17 (Table 4). Notably, only 54.35% (50/92) of participants were able to provide explicit details regarding EEHV prevention and control. Among the specific prevention practices for EEHV infection, the most frequently reported by mahouts were separation of new elephants from the existing herd (16.30%, 15/92), isolation of infected elephants from the rest of the group (15.22%, 14/92), and routine behavioral monitoring (14.13%, 13/92). The general practices encompassed four essential components: regular deworming, adherence to good husbandry practices (e.g., maintaining proper sanitation, rotating tie-stalls, and providing adequate shade), ensuring high-quality and sufficient food, and promoting ample exercise opportunities for the elephants.

Regarding elephant-elephant relations, the EKP elephant movement logbook revealed that approximately 30% (27/92) of mahouts took their elephant on outings for various purposes throughout the 2020–2021 period, neglecting the implementation of disease screening and quarantine protocols. These activities comprised group performances, breeding activities, religious ceremonies, live engagements on social media platforms, and gatherings within the village. Moreover, during COVID-19 pandemic, the international travel was banned, therefore affected to the elephant-based tourism, which altered the movement pattern of captive

**Table 3. Attitude score according to EEHV infection.**

| Attitude score | Total score | Mean ± SD | Range | number of participant's score ≥ mean (%) |
|---|---|---|---|---|
| 1. Elephant calves are the susceptible host | 2 | 1.6 ± 0.96 | 0–2 | 72 (78.26%) |
| 2. EEHV can cause fatality | 2 | 1.53 ± 0.81 | 0–2 | 68 (73.91%) |
| 3. EEHV can be treated | 2 | 1.13 ± 0.92 | 0–2 | 45 (48.91%) |
| 4. New coming elephants can shed the EEHV | 2 | 1.03 ± 0.91 | 0–2 | 39 (42.39%) |
| 5. Elephant can infect EEHV by contacting the EEHV survivors | 2 | 1.55 ± 0.79 | 0–2 | 68 (73.91%) |
| 6. Early weaning relate to EEHV infection | 2 | 0.98 ± 0.91 | 0–2 | 53 (57.61%) |
| 7. Vitamin C supplement is used for EEHV prevention | 2 | 1.22 ± 0.86 | 0–2 | 46 (50%) |
| 8. Chance of your elephant to get EEHV infection | 3 | 0.62 ± 0.82 | 0–2 | 41 (44.57%) |
| **Total** | 17 | 9.67 ± 4.28 | 0–2 | 59 (64.13%) |

**Table 4. Practicing score according to EEHV infection.**

| Practicing score | Total score | Mean ± SD | Range | number of participant's score ≥ mean (%) |
|---|---|---|---|---|
| Good management | 4 | 1.54 ± 0.84 | 0–4 | 47 (51.09%) |
| Specific EEHV prevention | 6 | 0.76 ± 0.82 | 0–3 | 50 (54.35%) |
| **Total** | 10 | 2.30 ± 1.17 | 0–5 | 38 (42.39%) |

elephants in Thailand. The EKP and Surin province was the hometown for large population of captive elephant that worked across country, thus during COVID-19, most of the elephants were moved back to these areas. The EKP gave a special allowance to non-registered elephants to resides within the EKP area during COVID-19. Data from elephant movement logbook showed 37 non-registered elephants were moved into the EKP areas in 2021. In addition, no disease screening or quarantine were performed for this group of elephants prior the entrance to the EKP areas. Contrary, although a number of non-registered elephants were moved back into the EKP areas, local activities, which normally attributed to the movement of EKP's elephants, was creased–leading to the low movement of registered elephant during COVID-19. Notably, despite being aware of the potential transmission of EEHV through direct elephant contact, a subset of mahouts (10.87%, 10/92) deliberately elected to visit and observe clinical EEHV-infected elephant.

For the elephant transportation, it was observed that proper cleanliness measures were lacking for both the elephants and the mahouts involved. In cases where elephant transport trucks were utilized, no cleaning process was performed. Conversely, when mahouts employed their personal lorries, it was found that 84% of them performed cleaning using water—however, only 38.46% utilized disinfectant agents such as dishwashing liquid and powdered detergent.

## Assessing the KAP scores in relation to mahout backgrounds

From the interviews, it was evident that everyone had distinctive attitudes and diverse experiences in elephant rearing. This analysis clarified the relationship between the mahouts' profiles and their KAP scores. According to Spearman's correlation analysis, there were low negative correlations between knowledge scores and the age of the mahouts, which were statistically significant ($p < 0.05$). The correlation coefficient ($r_s$) stood at -0.23 (Table 5). In addition, no significant correlations were observed between mahout's age and attitudes and practice's scores. Moreover, it was important to emphasize that the analysis revealed no significant differences in KAP scores when comparing different groups based on mahout experience. Additionally, there were no significant correlations found between the amount of mahout experience and their KAP scores.

Regarding the relationship between knowledge, attitudes, and practice scores, a significant positive correlation at a low level ($p < 0.05$) was found between the knowledge score of EEHV and the practice score. The correlation coefficient ($r_s$) was 0.35 (Table 6). The data used to generate the means, standard deviations, and comparative analyses for Tables 1–6 were presented in S1 Dataset.

**Table 5. Comparative analysis of KAP scores with age of mahout and elephant rearing experience.**

| Variable | Age of mahout | Elephant rearing experience |
|---|---|---|
| Knowledge score (K) | -0.23731** | -0.14981 |
| Attitude score (A) | -0.06849 | 0.09555 |
| Practice score (P) | 0.10377 | 0.1460 |

** significantly different at the 0.05 level.

**Table 6. The correlation coefficient represents the relationship between the participants' knowledge score, attitude, and practice regarding EEHV.**

| Variable | Knowledge score | Attitude score | Practice score |
|---|---|---|---|
| Knowledge score (K) | 1 | 0.03338 | 0.35231** |
| Attitude score (A) | | 1 | -0.07951 |
| Practice score (P) | | | 1 |

** significantly different at the 0.05 level.

When analyzing the data by dividing elephant owners into two groups based on the age of the elephants they cared for, the group caring for young elephants (under 10 years old) consisted of 51 individuals, while the group caring for adult elephants (10 years and above) consisted of 41 individuals. The results showed significant differences in KAP scores between the two groups in all three aspects (p < 0.05) using the two-samples Wilcoxon test. It was also observed that the group caring for young elephants had significantly higher average scores for knowledge and practices related to EEHV. However, in terms of attitudes, the group caring for adult elephants showed higher awareness of the severity of the disease and a greater agreement with factors influencing viral infection compared to the group caring for young elephants, which was a higher-risk group for EEHV infection and severe disease symptoms (Fig 3). The data used to generate Fig 3 was presented in S1 Dataset.

In comparing the differences among those with prior experience with EEHV, five interviewees had previously cared for elephants infected with EEHV. When analyzing the differences in KAP scores between the experienced and non-experienced groups using the Mann-Whitney U test, no statistically significant differences were found across all three aspects.

## Discussion

Thailand has a long-standing tradition of housing captive Asian elephants, and Surin province, situated in the country's northeast region, is particularly renowned for its captive elephant

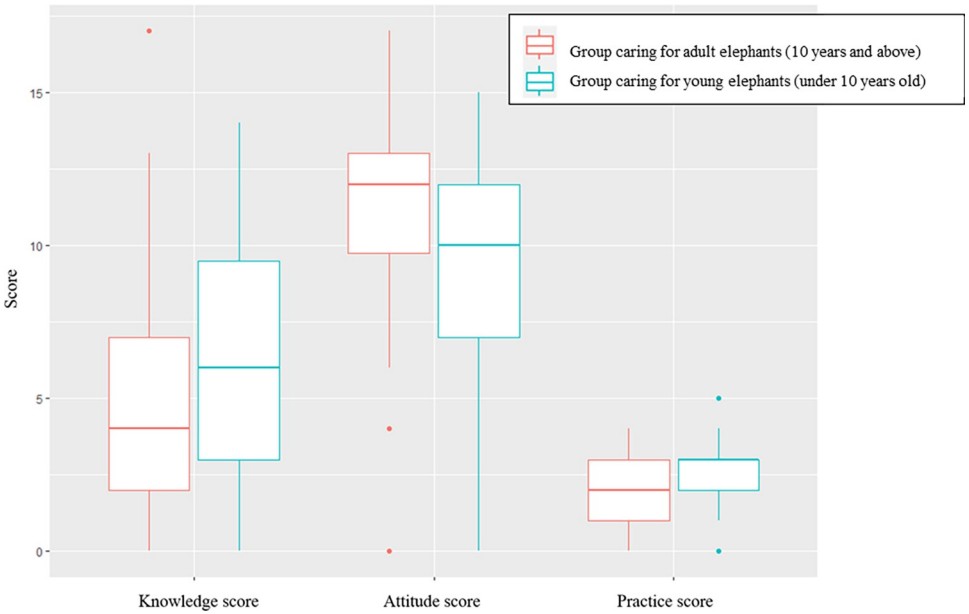

**Fig 3. The boxplot compared KAP scores of mahouts caring for adult elephants and those caring for young elephants.**

population. The mahouts, who serve as the primary caretakers of the elephants, develop strong bonds with their elephants. Previous research has emphasized the close interactions between mahouts and elephants, highlighting the mahouts' significant role as a valuable source of knowledge about captive Asian elephants [22]. The traditional practices associated with elephant care and handling have been handed down through generations, becoming part of the cultural and belief systems surrounding elephant rearing [7, 27]. It is essential to recognize that personal, cultural, and religious beliefs profoundly influence health practices, or called ethnoveterinary medicine [24, 28–30], which is also evident in this study on elephant care and treatment decision. For example, most of the mahouts in this study demonstrated a strong belief in the low risk of EEHV infection posed by their elephants. This belief likely stemmed from their personal and cultural practices, which allowed the elephants to freely browse and forage for herbs, barks, and trees during the day. The mahouts believed that elephants had the instinct to select plants that would benefit to their health. These findings are consistent with a study from Laos, where mahouts similarly allowed their elephants to feed in the forest on specific plants believed to improve elephant's health [31]. Moreover, interviews conducted with three mahouts whose elephants experienced clinical EEHV infections in 2017 revealed their belief in both traditional methods and modern medication. The elephants were treated at home because the mahouts believed that transporting the animals to the hospital would cause undue stress. However. antiviral drugs were given alongside spiritual practices performed by a local hermit. These practices included praying, drinking holy water, and spraying consecrated white liquor. This approach is widespread in cultures practicing captive elephant husbandry, highlighting the intricate relationship between humans and elephants they rear [32]. Although eleven EEHV clinical cases had been confirmed in this area previously [11, 12], none of the EEHV clinical cases were recorded during study period. However, twelve and two elephant calves were tested positive for EEHV without developing clinical signs in 2020 and 2021, respectively (Bangkeaw, personal communication).

Regarding the relationship between knowledge scores and disease symptoms, 27.17% of interviewed elephant owners mentioned facial swelling as the most prominent symptom specific to EEHV infection. This was followed by edema in the areas of the body (19.57%) and tongue cyanosis (17.39%). However, it should be noted that the initial stages of the disease do not exhibit specific symptoms. Previous research indicated a correlation between changes in sleeping behavior and depression during the early stages of the disease [12], which aligns with the findings on knowledge of disease symptoms in this study. The majority (34.78%) of elephant owners mentioned depression as a symptom, followed by loss of appetite (15.22%) and diarrhea (11.96%). Early treatment and timely care increase the chances of survival for infected elephants [12], consistent with the survival analysis of infected elephants, highlighting the importance of prompt treatment in improving survival rates [11]. The routine observation of any change in elephant symptoms was also emphasized in knowledge scores for preventing EEHV, with 33.70% of respondents mentioning it. However, there were misconceptions and confusion about the disease and its differentiation from other illnesses. For example, swelling of face and neck in elephants was sometimes mistaken to hemorrhagic septicemia, allergies, or obesity. This could easily be differentiated from EEHV through PCR or hematology tests. However, this misconception often led to delays in notifying veterinarians. Therefore, increasing knowledge about EEHV, including its symptoms, transmission, and preventive measures, is essential. Improved understanding among elephant owners/caretakers can lead to better practices related to the KAP model, which represents the interconnection among knowledge, attitudes, and practices that can lead to changes in individual behaviors [33–35]. This model demonstrated that knowledge and attitude influence practice, even if they are not necessarily correlated [36]. Additionally, improved understanding of EEHV among mahouts can lead to

better practices for disease prevention, which may not have been statistically correlated previously.

From the attitude score results, it was evident that even though over two-thirds of respondents were aware of the severity of the disease, their lack of understanding about the disease, coupled with their reliance on personal experiences to form their opinions, led to a generally negative attitude towards certain perceived risks of infection. For instance, regarding the risk of infecting with EEHV from new-coming elephants, respondents reasoned that the introduction of the disease by a new elephant should be apparent upon first sight; if an elephant appears healthy, it is unlikely to carry or spread the disease. Additionally, they argued that no one would bring a sick elephant into contact with others, as this would risk worsening the illness and spreading it further. However, respondents lack understanding of the concept of disease carriers, where an elephant can carry the virus without showing symptoms and can transmit it when the virus reactivates [37, 38]. Similarly, in the context of training early weaned elephants, respondents observed no clear connection between early weaning and the incidence of illness. Even among mahouts who regularly train young elephants, there was no perceived link between early weaning and the risk of EEHV infection. This is why they do not recognize the relationship between early weaning and the increased risk of infection.

The nature of herpesvirus, which could establish latency and subsequently reactivate, in humans and other animal species is directly related to stress levels and immune response [39, 40]. Stress can lower the immune response, making individuals susceptible to viral infection [12]. One of the causes of stress is movement or changes in habitat, such as introducing new elephants [41]. This study was conducted during the COVID-19 pandemic when tourist elephant camps were closed, and most of the captive elephants went back to their hometown, including returning to Surin. Results found that 41.67% (n = 10/24) of elephants were moved for various purposes, and 37% were brought into the area without a clear disease quarantine or screening measures. These practices can induce stress in the elephant and potentially introduce other infections such as tuberculosis into the project area [42, 43]. Additionally, age is a factor affecting the immune response to EEHV. Offspring receive maternal immunity through transplacental passive immunity, which decreases when elephants reach 24–36 months of age, indicating the period of susceptibility and manifestation of severe symptoms [44]. This study highlights that most elephant owners choose to wean elephants under two years old, coinciding with the decline of maternal immunity. It could be one of the factors contributing to illness and fatality in calving elephants in this area.

Lack of cleanliness measures during elephant transportation was noted, especially when shared trucks were used. Ethyl alcohol was reported to inhibit the function of herpesviruses [45–47], as well as the use of sodium linear alkyl benzene sulfonate (LAS) at a concentration of 0.0125% for 10 minutes, which can inhibit enveloped equine herpesvirus type 1. Other agents, such as 13% hydrogen peroxide for 5 minutes, a 0.2% benzalkonium chloride solution, and 0.12% chlorhexidine digluconate solutions have shown the ability to inhibit the herpes simplex virus [45]. Therefore, it is recommended that the EKP should provide cleaning stations for vehicles. The cleaning process should include sweeping away all waste, rinsing with water for at least 10 minutes, applying the cleaning agent, and then allowing it to dry to ensure the effectiveness of virus inhibition.

The analysis of the differences between the group of elephant owners caring for young elephants and the group caring for adult elephants in this study demonstrated differences in knowledge, attitudes, and practices. It found that the group caring for young elephants had higher average scores in knowledge and practices than those caring for adult elephants. One of the reasons for the previous public relations focus on disseminating information on EEHV is primarily targeting the group of elephant owners caring for young elephants. However,

elephants of all age groups can become infected with EEHV and exhibit different clinical symptoms. Infected elephants that survived can also be carriers and transmit the virus to other elephants [37, 38]. Therefore, providing comprehensive knowledge and information about EEHV to all elephant owners is crucial.

This study findings highlighted that the majority of mahouts have strong confidence in their traditional practices, firmly believing that these practices contribute to the overall health and well-being of their elephants, thereby reducing the risk of infection by EEHV. These practices encompass using herbal remedies for disease prevention and treatment, applying Thai rice whisky for pain relief and protection against evil spirits, and providing opportunities for elephants to freely explore and feed on herbs and trees to support their recovery. These deeply rooted beliefs lead mahouts to prioritize traditional methods over scientific approaches, such as good hygiene management and specific prevention strategies for EEHV. As a result, practice scores related to modern disease prevention were relatively low. This study's results revealed a correlation between knowledge scores and practice scores, suggesting that limited exposure to current disease information and scientific advancements contributes to these lower scores. In many traditional communities, long-established cultural practices are highly valued and strictly followed, making it quite challenging to introduce new information or modern medicine to the community. The understanding and diagnosis of EEHV have evolved significantly over the past two decades, potentially making it challenging to introduce modern disease management strategies and scientific knowledge to the traditional elephant-raising culture. Hence, when disseminating information about new diseases such as EEHV within long-standing communities like Surin, it becomes imperative to integrate this newfound knowledge with their existing beliefs and traditions. One approach is to demonstrate respect for local culture while also building trust and openness to imported knowledge and treatment methods [48]. Additionally, it is important to emphasize that combining traditional remedies with modern medicine can enhance treatment coverage and efficacy [49, 50], maximizing benefits for the animals. Furthermore, it is recommended to apply the KAP survey in different elephant-handling cultures to understand their traditional practices and explore ways to integrate modern knowledge into their communities.

## Conclusion

This study indicated that mahouts participating in the Elephant Kingdom Project demonstrated a positive attitude and awareness toward EEHV infections. However, there needs to be more knowledgeable and preventive practices concerning EEHV. Therefore, it is imperative to urgently implement public education initiatives to provide accurate and up-to-date information about EEHV to the community. These initiatives emphasize the importance of early detection of behavioral changes, symptom surveillance, and disease prevention practices. By enhancing knowledge levels, disease awareness can be increased, leading to the development and implementation of effective disease prevention and control measures. Furthermore, a tailored public communication plan should be developed to suit the community's needs, and ongoing data analysis should be conducted to assess improvements over time. Additionally, given the frequent gatherings of elephants in the area and past cases of EEHV infection, a risk analysis should be conducted to evaluate the level of disease transmission risk, thereby guiding the development of disease prevention strategies aligned with official disease announcements.

## Supporting information

**S1 Dataset. Minimal data.**
(DOCX)

## Acknowledgments

We sincerely thank the Zoological Park Organization of Thailand for permitting us to collect data within the Elephant Kingdom Project. We also express our heartfelt gratitude to all the participants and dedicated staff members involved in the Elephant Kingdom Project. Their cooperation, interviews, data provision, and invaluable assistance throughout the data collection process were essential to the success of our research.

## Author Contributions

**Formal analysis:** Narueporn Kittisirikul.

**Investigation:** Narueporn Kittisirikul.

**Methodology:** Narueporn Kittisirikul.

**Resources:** Nuttapon Bangkaew.

**Supervision:** Waraphon Phimpraphai, Supaphen Sripiboon.

**Writing – original draft:** Narueporn Kittisirikul.

**Writing – review & editing:** Waraphon Phimpraphai, Supaphen Sripiboon.

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
