## [Decision Letter · Decision Letter 0]

26 Apr 2024

PONE-D-23-38946Unlocking Insights: Mahout’s Perceptions and Practices in Managing EEHV infection Among Captive Asian Elephants in Surin province, Thailand.PLOS ONE

Dear Dr. Sripiboon,

Thank you for submitting your manuscript to PLOS ONE. After careful consideration, we feel that it has merit but does not fully meet PLOS ONE’s publication criteria as it currently stands. Therefore, we invite you to submit a revised version of the manuscript that addresses the points raised during the review process. More precisely, according to the comments of the three reviewers, many aspects should be reconsidered and this study would be deeply reframed. I particularly agree with comments of reviewer 1, who suggest to propose more explicit hypothesis, e.g. comparing different stakeholders/areas?), to give additional data collection (e.g. key-informant interviews? History of EEHV cases diagnosed/suspected for each mahout), and to deeply reanalyse the data.

We look forward to receiving your revised manuscript.

Kind regards,

Johan R. Michaux

Academic Editor

PLOS ONE

Journal Requirements:

2. Thank you for stating the following financial disclosure: "This research benefited from the support of two funding sources. Narueporn Kittisirikul received the Kasetsart Veterinary Development Funds grant (number 64_03), which was employed for both data collection and publication purposes. Furthermore, she obtained partial funding from the Faculty of Veterinary Medicine, Kasetsart University (grant number 2563-1/02), which assisted in covering master's degree tuition fees and providing salary support."

3. Thank you for stating the following in the Acknowledgments Section of your manuscript: "We sincerely thank the Zoological Park Organization of Thailand for permitting us to collect data within the Elephant Kingdom Project. We also express our heartfelt gratitude to all the participants and dedicated staff members involved in the Elephant Kingdom Project. Their 20/23 cooperation, interviews, data provision, and invaluable assistance throughout the data collection process were essential to the success of our research. Lastly, we express our appreciation to the Kasetsart Veterinary Development Funds for their financial support of the research project. Additionally, this work was partially funded by the Faculty of Veterinary Medicine, Kasetsart University."

Please remove any funding-related text from the manuscript and let us know how you would like to update your Funding Statement. Currently, your Funding Statement reads as follows: "This research benefited from the support of two funding sources. Narueporn Kittisirikul received the Kasetsart Veterinary Development Funds grant (number 64_03), which was employed for both data collection and publication purposes. Furthermore, she obtained partial funding from the Faculty of Veterinary Medicine, Kasetsart University (grant number 2563-1/02), which assisted in covering master's degree tuition fees and providing salary support."

Reviewers' comments:

Reviewer's Responses to Questions

**Comments to the Author**

1. Is the manuscript technically sound, and do the data support the conclusions?

Reviewer #1: Partly

Reviewer #2: Yes

Reviewer #3: Yes

2. Has the statistical analysis been performed appropriately and rigorously? 

Reviewer #1: I Don't Know

Reviewer #2: Yes

Reviewer #3: Yes

3. Have the authors made all data underlying the findings in their manuscript fully available?

Reviewer #1: Yes

Reviewer #2: Yes

Reviewer #3: Yes

4. Is the manuscript presented in an intelligible fashion and written in standard English?

Reviewer #1: Yes

Reviewer #2: Yes

Reviewer #3: Yes

5. Review Comments to the Author

Reviewer #1: General comment:

I have reviewed the paper entitled “Unlocking Insights: Mahout’s Perceptions and Practices in Managing EEHV infection Among Captive Asian Elephants in Surin province, Thailand.”, submitted by Sripiboon et al. for publication in PLOS One.

The paper addresses the topic of perceptions and practices of a group of stakeholders regarding an emerging disease affecting captive elephants in Thailand. This topic could have led to very interesting insights in the field of participatory veterinary epidemiology, possibly exploring original facets of Man-Nature relationships, domestication of wild mammals, efficiency of “traditional management practices” against new emerging infectious diseases. Unfortunately, as stated at the end of the introduction, the focus of the paper is in fact narrow, focusing on one group of stakeholders (‘mahouts’) from a limited area (EK project/Thea Tum District/Surin province), and on a specific disease (EEHV). The study relied on “top-down approach” using a “knowledge, attitudes and practices (KAP) survey”, which demonstrated (inevitably) that mahouts had imperfect or inappropriate knowledge/attitudes/practices as compared to the conventional veterinary knowledge, taken as the absolute reference despite the fact that major gaps remain on the knowledge and management of emerging EEHV. The fact that the authors do not account for uncertainties regarding the knowledge systems, and because the survey was not framed with a comparative approach or based on clear hypothesis (e.g. do mahouts know more/behave better/… than other stakeholders, and/or in other provinces, and/or for other non emerging diseases…) the inevitable conclusion reached is that they need to be better informed and educated by veterinarians. On the other hand, the paper does not provided sufficiently detailed epidemiological data regarding EEHV cases, and other diseases affecting elephants in Surin/other provinces, in other districts, and most importantly the number of elephants diagnosed with EEHV for each of the 96 mahouts surveyed?

I therefore believe that the paper is not acceptable in its current state for publication in PLOS One. I’m afraid that the necessary revisions may be out-ot-reach, as they would require a complete reframing of the paper (i.e. based on more explicit hypothesis, e.g. comparing different stakeholders/areas?), additional data collection (e.g. key-informant interviews? History of EEHV cases diagnosed/suspected for each mahout), and reanalyzing the data. I have listed below some suggestions of edits/corrections that the authors may want to consider if they the editor invites them to resubmit to this journal, or if they decide to submit to another journal.

Specific comments and editorial suggestions:

- Title: suggest to remove “Unlocking insights:”; not clear which insights are referred to, and whether they are unlocked by the survey?…

- Abstract: Line 32: unclear what “disease information” means

- L37: do you mean “contract EEHV”?

- L70: change sentence “ … medicine, the threat of elephant endotheliotropic… was recently prioritized. EEHC is a fatal hemorrhagic… documented in Thailand…”

-L73: consider changing “carries” into “induces/is responsible for”

-L78: add “years” after “1 and 8”

-L82: change “promising” into “conclusive” or “ efficient”

-L93: repeat “specific”

-L97: do you mean “extent” instead of “impact”

-L107: change “the survey” to “This survey” or “Our survey”

-L110: add “activities” after “…elephant tourism activities”

- L122: correct “…area presented…

-L124/FiLegend: Correct “Map of “ instead of “Mapping…”

-L125/FiLegend: Correct “sampled “ instead of “sampling…”

-L126/FiLegend: remove “The display…. QGIS [22]” from the figure legend and possibly add to Mat&Meth

-L131:correct “semi-srtructured”

-L131: clarify “observations” of what?

-L133: give n= mahouts with elephants <10 years

- L153, and whole section Mat & Metho: clarify why the number of “points”/give different weights is different between subjects?

- L153: suggestion to add a sensitivity analysis clarifying the impact of the subjective weighting/subject on the end-results and conclusions

- L182/Statistical analysis: remove unnecessary reference to Excel

- L186/clarify which stat. test were used to compared which variables/groups

- L190: clarify how many mahouts in total are involved in EKP

-L196: unclear by whom the system is known, and maybe indicate the thai name of the system (if it’s a translation)?

-Fig2, Fig3, Fig4 : remove the pictures, which are not very useful, especially 3 and 4

-L211-2: Clarify semantics: which language was used, which term in vernacular language used,…

-L211: what question was asked to the mahouts exactly to assess if the respondents knew about EEHV?

-L217/Table 2 title: replace “according” by “regarding”?

-Table 2: Why no question was asked about potential Treatment of EEHV? Besides current veterinary knowledge, maybe some respondents may have some ideas/experience of treatments (e.g. symptomatic treatements?)

-L219: change”the study” into “our study”

-L225: what was exactly the question asked regarding the transmission route? Did you leave the question open for possible (local knowledge) alternative etiologies?

- L238: consider replacing “concerning” by “associated with newly…”

Table3: correct “1. Calve elephant…”

Revise unclear “5. Infection EEHV from infected..;”?

-L248: clarify what are “the concerns…”

-L247-8: revise: an elephant can’t “roam freely” if it is chained

-L247: revise “under natural elements”

-L265: provide “evidence”: what data did collect providing evidence?

-L268: clarify how Covid-19 restrictions/lock-downs affected elephant movements and what evidence there is that it altered the contac patterns between elephants, and EEHV spread?

-L275: clarify “… the claning process needed to be specified..”

-L280: revise title”the relation…”

-L281-4: unclear 1st sentence: remove?

-L291: revise title “Correlations between KAP scores…”

-L3030-4, and throughout the next: specify “knowledge and practices related to EEHV”

-L313: Revise “knowledge scores of mahouts caring for adult elephants and those caring for young elephants”

-L323: add reference or anthropological or historical records reporting how and by whom “traditional pratices… handed down through generations”

-L326: provide data supporting the allegation that cultural and religious beliefs influence…”; no evidence provided with the KAP data presented

-L340: give precise example of “other illnesses”, and differential diagnostic?

-L343: clarify what is “KAP model”?

-L343-4: this sentence is puzzling… why did you calculate the correlation coefficients if “they are not necessarily correlated”?

-L345: clarify “nature of herpesvirus”?

-L353: indicated which “other infections” exactly

-L32: add “risk of infection by EEHV”

-L375: “clarify where and when “elephants freely explore…”and give evidence/quote mahouts saying “herbs and and tres support their recovery (from what/how?)

-L377-8: clarify why they would be marginalized from their community/other mahouts?

-L383: provide indications how you intend to achieve this challenging goal; “integrate newly found knowledge…”

-Bibliography: I did not check in details the reference/formats but noticed that it is not consistent throughout (name of journals in full/abbreviated, missing DOIs, missing page numbers/volume etc ….)

-Map: unclear why the authors included this map with no indication of Thailand/Surin province, and no mention in the text regarding the details indicated streets, villages, etc…

-Pictures: remove? Unclear why the authors included these (except maybe picture

pictures1 illustrating contacts between two elephants (and potential spread of EEHV?)

Reviewer #2: This study would benefit from additional analysis to strengthen its research findings and derive more insights from the data. Specifically, the study should investigate the correlation between demographic data and Knowledge, Attitude, and Practices (KAP). For example:

1. Age and KAP: The current age range of 15–60 years is quite broad. It would be beneficial to recategorize age groups for a more detailed comparison.

2. Number of elephants and KAP: Similarly, the range of 2-5 elephants is too wide. Recategorizing this variable could provide more nuanced insights.

3. Experience level and KAP: Recategorizing the experience variable and conducting correlation analysis could reveal valuable relationships.

These adjustments could enhance the study's findings and provide a more nuanced understanding of the factors influencing KAP related to elephants.

Reviewer #3: Comments

This study used the knowledge, attitude, and practice (KAP study) toward EEHV management via 92 mahouts’ interview, and showed the lower knowledge and practice scores than the expectations, indicating an insufficient understanding of disease information and prevention.

The study was important with the traditional of mahout management in caring elephant, related to the EEHV occurrence, which could be applied in other parts of countries, and Asian elephant range countries.

Topic

EEHV is not generally recognized by readers, mostly by elephant people only; thus, please add the full name with abbreviation in bracket.

Introduction

As there were reports and epidemiology of the EEHV occurrence in Thailand (Boonprasert et al. 2019, Yun et al., 2021), please mentioned or showed the previous case number of EEHV cases in the elephant kingdom project.

Line 30

Please add “(KAP score)”

Line 106

Knowledge, Attitudes, and Practices should be changed as KAP only, as it was repeated.

Material and mthods

Line 174

32 should be changed to 31

Line 118-119, 191

Please use only “EKP”, as the full name was mentioned in line 62

Results

The score of each KAP categories was very low (Knowledge 3.78 from 30; Attitude 9.67 from 17; Practice 2.3 from 30). Please discuss more in the discussion part.

- What is the interpretation of each categories?

- Is that much differ from the KAP score in other previous study of other infectious disease e.g. COVID-19?

Line 213

“ranging from 0 to 11” and refer to Table 2, which showed the range of “0-17”. I am not sure which one is correct.

Line 273-278

The authors showed the results of transportation with cleanliness. Please discuss more in the discussion part.

Line 386-389

The author mentioned “recommendation to apply the KAP survey in other parts of the country” in the abstract, but no mentioned in the end of the discussion part or at the conclusion part. Please add this part.

Table 2 and Table 3

There were five columns in Table 2 and 4, and four columns in Table 3.

The different was the “range” column; therefore, this should be consistent.

General comment

Line 72, 78, 359

The author used “Calf elephant”. In my opinion, it’s better used “elephant calf”, which is similar to several previous arti

6. PLOS authors have the option to publish the peer review history of their article (what does this mean?). If published, this will include your full peer review and any attached files.

Reviewer #1: No

Reviewer #2: **Yes: **Assoc Prof Dr Tuempong Wongtawan DVM MS MVM PhD SFHEA

Reviewer #3: No

---

## [Author Response · Author response to Decision Letter 0]

20 Jun 2024

1. Please ensure that your manuscript meets PLOS ONE’s style requirements. 

o The authors had checked throughout the manuscript the ensure that it meets PLOS ONE’s style requirements. 

2. Please state what role of the funders took in this study.

o The funders had no role in study design, data collection and analysis, decision to publish, or preparation of the manuscript. 

3. Please remove any funding-related text from the manuscript. 

o The funding-related text had been removed from the acknowledgement in the manuscript. 

4. The copyright issue of Figure 1. 

o Figure 1 had been removed from the manuscript.

Reviewer1:

General comments 

Reviewer 1 has recommended a complete reframing of the paper by comparing different stakeholders/areas or adding more data collection and reanalyzing the data. 

o Response in Page 4, line 84-85, Page 23, line 446-448: due to the limited of funding and study timeframe, the authors were unable to compare the results with other stakeholders/areas, however, Surin province is the place where most of the baby elephants were born and been transported to other area of Thailand. Therefore, it counts as a biggest population of captive elephant in Thailand with unique traditional management system. Beginning the study in Surin province could provide a database for captive elephant population in Thailand. In addition, the authors have added information on the history of EEHV diagnosed/suspected cases in Surin and also reanalyze some data, as reviewer suggested (see details below).

Specific comments and editorial suggestions

- Title: suggest removing “Unlocking insights:”, not clear which insights are referred to, and whether they are unlocked by the survey?

o Response in Title (page 1): the authors would like to remain “Unlocking insight”, if possible. As it refers to the in-depth personal data collection from elephant keepers which has not been extensively explored before in Thailand. However, if the editorial team agreed on removing ‘unlocking insights’, the authors are also fine to remove.

- Abstract: Line 32: unclear what “disease information” means 

o Response in Page 2, line 33-34: change to “nature of disease and preventive measures” to clarify the “disease information”.

- Abstract: Line 37: do you mean “contract EEHV”?

o Response in Page 2, line 39: changed to “receive”.

- L70: change sentence “… medicine, the threat of elephant endotheliotropic…was recently prioritized. EEHV is a fatal hemorrhagic … documented in Thailand ….”

o Response in Page 4, line 81-83: changed as reviewer suggested. 

- L73: consider changing “carries” into “induces/is responsible for”

o Response in Page 5, line 86: changed to “is responsible for”.

- L78: add “years” after “1 and 8” 

o Response in Page 5, line 93: added “years” as reviewer suggested. 

- L82: change “promising” into “conclusive” or efficient

o Response in Page 5, line 97: changed to “conclusive”.

- L93: repeat “specific”

o Response in Page 6, line 109: changed to “key”.

- L97: do you mean “extent” instead of “impact”

o Response in Page 6, line 114: changed to “extent”.

- L107: change “the survey” to “This survey” or “Our survey”

o Response in Page 6, line 126: changed to “This survey”.

- L110: add “activities” after “… elephant tourism activities” 

o Response in Page 6, line 129: added “activities” as reviewer suggested.

- L122: correct “… area presented …” 

o Figure 1 (the map) was removed.

- L124/Figure legend: correct “Map of” instead of “Mapping”

o Figure 1 (the map) was removed.

- L125/Figure legend: correct “sampled” instead of “sampling”

o Figure 1 (the map) was removed.

- L126/Figure legend: remove “The display .. QGIS [22]” from the figure legend and possibly add to Mat&Meth

o Figure 1 (the map) was removed.

- L131: correct “semi-structured”

o Response in Page 8, line 149: changed to “semi-structured”. 

- L131: clarify “observations” of what?

o Response in Page 8, line 149: changed “observation method” to “field observation.”

o Response in Page 9, line 172-174: added the definition of field observation “In addition, the field observation was applied to assess husbandry, waste management, and general care provided to elephants, in comparison to insights gathered through interviews”.

- L133: give n = mahouts with elephant < 10 years

o Response in Page 8, line 152: added the number of mahouts (n=41).

- L153: and whole section Mat&Metho: clarify why the number of “points”/give different weights is different between subjects?

o Response in Section “Data collection method” (Page 8-10): rewritten and added relevant information as reviewer suggested.

- L153: suggestion to add a sensitivity analysis clarifying the impact of the subjective weighting/subject on the end-results and conclusions

o Response in section “Assessing the KAP scores in relation to mahout backgrounds” (Page 19-22): In this study, sensitivity analysis was conducted using correlation analysis test and Wilcoxon, and the Mann-Whitney U Test 

- L182/Statistical analysis: remove unnecessary reference to Excel

o Response in Page 11, line 219: removed Microsoft Excel 

- L186: clarify which stat test were used to compared which variables/group

o Response in Page 11, line 221-228: added “The statistical analysis included correlation testing to examine the relationships between knowledge, attitude, and practice scores, and the mahouts' demographics (such as age and experience in elephant handling). Additionally, the Wilcoxon test was used to compare KAP scores between mahouts who care for elephant calves and those who care for adult elephants. While the Mann-Whitney U test was used to compare KAP scores between the experienced in EEHV infected elephant groups and the non-experience groups.”

- L190: clarify how many mahouts in total are involved in EKP

o Response in Page 11, line 232-233: added “From the total of 200 mahouts in EKP, this study included 92 participants (mahouts) from 81 households”.

- L196: unclear by whom the system is known, and maybe indicate the Thai name of the system (if it’s a translation)?

o Response in Page 12, line 237-240: removed one-by-one system to prevent confusion and added data regarding to the elephant management in this study area. 

- Fig2, Fig3, Fig4: remove the pictures, which are not very useful, especially 3 and 4

o Response in Page 12, line 249-252: the authors would like to remain the pictures as they were illustrated the elephant husbandry in this study areas, which probably useful for the reader. However, the authors combine Fig 3 and 4 into single picture to present the resting area of elephant in this study. 

- L211-2: Clarify semantics: which language was used, which term in vernacular language used…

o Response in Page 14, line 262-270: rewritten and added more data for clarification. 

- L211: What question was asked to the mahouts exactly to assess if the respondents knew about EEHV?

o Response in Page 14, line 261-262: The interview was begun by directly asking "Do you know herpesvirus in elephants?".

- L217/Table 2 title: replace “according” by “regarding”?

o Response in Page 14 (Table 2): changed to “regarding”.

- L217/Table 2: Why no question was asked about potential treatment of EEHV? Besides current veterinary knowledge, maybe some respondents may have some ideas/experiences of treatment (e.g. symptomatic treatment?)

o The participants were asked about potential treatment of EEHV and their experience in EEHV treatment, however, the data was not showed in the original manuscript. The data regarding to EEHV treatment was added in the revision (see Page 23, line 436-443): “Moreover, interviews conducted with three mahouts whose elephants experienced clinical EEHV infections in 2017 revealed their belief in both traditional methods and modern medication. The elephants were treated at home because the mahouts believed that transporting the animals to the hospital would cause undue stress. However. antiviral drugs were given alongside spiritual practices performed by a local hermit. These practices included praying, drinking holy water, and spraying consecrated white liquor”.

- L219: change “the study” into “our study”

o Response in Page 15, line 278: changed to “this study” for consistency throughout the document. 

- L225: what was exactly the question asked regarding the transmission route? Did you leave the question open for possible (local knowledge) alternative etiologies?

o Yes, the question was open questions about how EEHV was transmitted? The final question was a close question to confirm the knowledge of main transmission route, the final question was “EEHV can be transmitted from elephant to elephant, yes or no?”. Moreover, additional data was added to clarify the knowledge on transmission (see Page 15, line 287-293).

- L238: consider replacing “concerning” by “associated with newly”

o Response in Page 16, line 310: changed to “associated with newly”. 

- Table 3: correct “1. Calve elephant” and revise unclear “5. Infection EEHV from infected?

o Response in Table 3 (Page 17): the misspelling was corrected, and the attitude subject was rewritten to make it clearer. 

- L244: clarify what are “the concerns”

o Response in Page 16, line 314: added “which could interfere protective immune development”.

- L247-8: revise: an elephant can’t “roam freely” if it is chained

o Response in Page 16, line 318: rewritten to “freely browse for food in the forested area with a long chain”.

- L247: revise “under natural elements”

o Response in Page 17, line 318: removed “under natural elements” to prevent the confusion.

- L265: provide “evidence”: what data collect providing evidence?

o Response in Page 18, line 341-342: added “the EKP elephant movement logbook”.

- L268: clarify how Covid-19 restrictions/lock-downs affected elephant movements and what evidence there is that it altered the contact patterns between elephants, and EEHV spread?

o Response in Page 18-19, line 347-359: added the effect of COVID-19 on elephant movement. 

- L275: clarify “... the cleaning process needed to be specified”. 

o Response in Page 19, line 365-366: rewritten and changed to “no cleaning process was performed”.

- L280: revise title “the relation…”

o Response in Page 19, line 370: changed the subheading to “Assessing the KAP score in relation to mahout backgrounds”.

- L281-4: unclear 1st sentence: remove?

o Response in Page 19, line 371: removed and rewritten.

- L291: revise title “correlation between KAP scores”

o Response in Page 20, line 383 (Table 5): changed to “Comparative analysis of KAP scores with age of mahout and elephant rearing experience”.

- L303-4: and throughout the next: specify “knowledge and practices related to EEHV” 

o Response in Page 21, line 402: added “related to EEHV”.

- L313: revise “knowledge scores of mahouts caring for adult elephants and those caring for young elephants

o Response in Figure 3 title (Page 21): changed Fig 5 to Fig 3 and changed title to “The boxplot compared KAP scores of mahouts caring for adult elephants and those caring for young elephants”. 

- L323: add reference or anthropological and historical records reporting how and by whom “traditional practices” handed down through generations”

o Response in Page 22, line 428: added the reference.

- L325: provide data supporting the allegation that cultural and religious beliefs influence…” no evidence provided with the KAP data presented

o Response in Page 22-23, line 426 – 449: added supporting data as reviewer suggested. 

- L340: give precise example of “other illnesses” and differential diagnostic?

o Response in Page 24, line 466 - 469: added “For example, swelling of face and neck in elephants was sometimes mistaken to hemorrhagic septicemia, allergies, or obesity. This could easily be differentiated from EEHV through PCR or hematology tests. However, this misconception often led to delays in notifying veterinarians.”.

- L343: clarify what is “KAP model”

o Response in Page 24, line 472 - 474: added “KAP model, which represents the interconnection among knowledge, attitudes, and practices that can lead to changes in individual behaviors”.

- L343-4: this sentence is puzzling .. why did you calculate the correlation coefficients if they are not necessarily correlated?

o Response in Page 24-25, line 474- 478: added “This model demonstrated that knowledge and attitude influence practice, even if they are not necessarily correlated. Additionally, improved understanding of EEHV among mahouts can lead to better practices for disease prevention, which may not have been statistically correlated previously.”

- L345: clarify “nature of herpesvirus”

o Response in Page 26, 497 - 499: added “The nature of herpesvirus, which could establish latency and subsequently reactivate, in humans and other animal species is directly related to stress levels and immune response”.

- L353: indicated which “other infection” exactly

o Response in Page 26, line 507 - 508: added “such as tuberculosis into the project area”.

- L372: add “risk of infection by EEHV”

o Response in Page 28, line 542: added “risk of infection by EEHV”.

- L375: clarify where and when “elephant freely explore..” and give evidence/quote mahouts saying “herbs and trees support their recovery (from what/how)?

o Response in Page 22-23, line 426-436: added more information on traditional belief of mahouts in the study area.

- L377-8: clarify why they would be marginalized from their community/other mahouts?

o Response in Page 28, line 551-554: added “In many traditional communities, long-established cultural practices are highly valued and strictly followed, which quite challenge to bring new information or modern medicine to the community.”

- L393: provide indications how you intend to achieve this challenging goal; “integrate newly found knowledge…”

o Response in Page 29, line 560-565: added data as reviewer suggested.

- Bibliography: the format was not consistent throughout 

o Response in Bibliography (Page 30-38): corrected the format of bibliography.

- Map: unclear 

o Removed the map (Figure 1)

- Pictures: remove? 

o Remain the figures as they represented the management in the study areas but combined them into one figure. 

Reviewer2:

1. Age and KAP: The current age of respondent range of 15-60 years is quite broad. It would be beneficial to recategorize age groups for a more detailed comparison.

o Response in Table 1 (Page 13): recategorized the age group of respondents to five groups, including ≤ 15, 16-30, 31-45, 46-60, and >60, indicating that 48% of respondents were age 31 – 45 years old. 

2. Number of elephants and KAP: similarly, the range of 2-5 elephants is too wide. Recategorizing this variable could provide more nuanced insights.

o Response in Table 1 (Page 13): recategorized the number of elephants under care to four groups, including 1, 2-3, 4-5, and > 5, indicating that 42% of respondents cared for 2-3 elephants. 

3. Experience level and KAP: recategorizing the experience variable and conducting correlation analysis could reveal valuable relationship. 

o Response in Table 1 (Page 13): recategorized the experiences with elephants to eight groups, including ≤ 5, 6-10, 11-20, 21-30, 31-40, 41-50, 51-60, and >60, indicating that most of the respondents had 11-40 years of experiences with elephants. 

o Response in Page 19, line 378-381: However, there was no significant difference between the experience of mahouts toward the knowledge, attitude, and practices of mahouts even recategorized.

Reviewer3:

- Topic: EEHV is not generally recognized by readers, mostly by elephant people only, thus please add the full name with abbreviation in bracket

o Response in Title (Page 1): changed to “Mahout’s Perceptions and Practices in Managing Elephant Endotheliotropic Herpesvirus (EEHV) infection Among Captive Asian Elephants in Surin province, Thailand.”

- Introduction: As there were reports and epidemiology of the EEHV occurrence in Thailand (Boonprasert et al., 2019, Yun et al., 2021), please mentioned or showed the previous case number of EEHV cases in the elephant kingdom project

o Response in Page 5, line 84-85: added “Addition, a total of 11 confirmed EEHV cases was reported in Surin with 66.7% fatality rate

---

## [Decision Letter · Decision Letter 1]

2 Sep 2024

Unlocking Insights: Mahout’s Perceptions and Practices in Managing Elephant Endotheliotropic Herpesvirus (EEHV) infection Among Captive Asian Elephants in Surin province, Thailand.

PONE-D-23-38946R1

Dear Dr. Sripiboon,

We’re pleased to inform you that your manuscript has been judged scientifically suitable for publication and will be formally accepted for publication once it meets all outstanding technical requirements.

Kind regards,

Cord M. Brundage, D.V.M., Ph.D.

Academic Editor

PLOS ONE

**Comments to the Author**

1. If the authors have adequately addressed your comments raised in a previous round of review and you feel that this manuscript is now acceptable for publication, you may indicate that here to bypass the “Comments to the Author” section, enter your conflict of interest statement in the “Confidential to Editor” section, and submit your "Accept" recommendation.

Reviewer #2: All comments have been addressed

Reviewer #3: All comments have been addressed

2. Is the manuscript technically sound, and do the data support the conclusions?

Reviewer #2: Yes

Reviewer #3: Yes

3. Has the statistical analysis been performed appropriately and rigorously? 

Reviewer #2: Yes

Reviewer #3: Yes

4. Have the authors made all data underlying the findings in their manuscript fully available?

Reviewer #2: Yes

Reviewer #3: Yes

5. Is the manuscript presented in an intelligible fashion and written in standard English?

Reviewer #2: Yes

Reviewer #3: Yes

6. Review Comments to the Author

Reviewer #2: (No Response)

Reviewer #3: This study used the knowledge, attitude, and practice (KAP study) toward EEHV management via 92 mahouts’ interview, and showed the lower knowledge and practice scores than the expectations, indicating an insufficient understanding of disease information and prevention.

The study was important with the traditional of mahout management in caring elephant, related to the EEHV occurrence, which could be applied in other parts of countries, and Asian elephant range countries.

The authors have responded all the comments very well.

However, there is small typing error, Please check

Line 447

Typing error: Change "claves" to "calves"

7. PLOS authors have the option to publish the peer review history of their article (what does this mean?). If published, this will include your full peer review and any attached files.

Reviewer #2: **Yes: **Tuempong Wongtawan

Reviewer #3: **Yes: **Chatchote Thitaram Faculty of Veterinary Medicine, Chiang Mai University

---

## [Editor Report · Acceptance letter]

1 Nov 2024

PONE-D-23-38946R1 

PLOS ONE

Dear Dr. Sripiboon, 

I'm pleased to inform you that your manuscript has been deemed suitable for publication in PLOS ONE. Congratulations! Your manuscript is now being handed over to our production team.

Kind regards, 

on behalf of

Dr. Cord M. Brundage 

Academic Editor

PLOS ONE